# Autophagy in Plant Abiotic Stress Management

**DOI:** 10.3390/ijms22084075

**Published:** 2021-04-15

**Authors:** Hong Chen, Jiangli Dong, Tao Wang

**Affiliations:** 1College of Grassland Science and Technology, China Agricultural University, Beijing 100193, China; chenhong115@126.com; 2State Key Laboratory of Agrobiotechnology, College of Biological Sciences, China Agricultural University, Beijing 100193, China

**Keywords:** autophagy, abiotic stress, autophagy-related genes, selective autophagy

## Abstract

Plants can be considered an open system. Throughout their life cycle, plants need to exchange material, energy and information with the outside world. To improve their survival and complete their life cycle, plants have developed sophisticated mechanisms to maintain cellular homeostasis during development and in response to environmental changes. Autophagy is an evolutionarily conserved self-degradative process that occurs ubiquitously in all eukaryotic cells and plays many physiological roles in maintaining cellular homeostasis. In recent years, an increasing number of studies have shown that autophagy can be induced not only by starvation but also as a cellular response to various abiotic stresses, including oxidative, salt, drought, cold and heat stresses. This review focuses mainly on the role of autophagy in plant abiotic stress management.

## 1. Introduction

Due to their sessile nature, plants are constantly exposed to many transient but recurring stresses. To improve their survival and complete their life cycle, plants have developed sophisticated mechanisms to maintain cellular homeostasis during development and in response to environmental changes [1,2]. Autophagy (which means "eating itself") is one of the most important homeostasis regulation systems for degradation and recycling of proteins and cytoplasmic organelles in plants throughout their life cycle [3]. Under normal conditions, basal autophagy functions as a housekeeping process to clear unwanted cytoplasmic contents and remobilize nutrients during leaf senescence and seed germination, whereas under certain stresses (starvation, oxidative, salt, drought, heat, cold, pathogen or fungal infection, and programmed cell death (PCD)), autophagy is up-regulated and contributes to the recycling of damaged or unwanted cell materials [4].

Several types of autophagy have been characterized to date, including macroautophagy, microautophagy, chaperone-mediated autophagy and organelle-specific autophagy [3]. The best-studied type of autophagy in plants is macroautophagy, in which autophagosomes form and then fuse with lysosomes or vacuoles to degrade cellular components and organelles [5]. In microautophagy, tonoplast or lysosome membrane invagination packages cytoplasmic components to form single-membrane autophagic bodies, and the substrate components are then released into the vacuolar lumen or lysosome. Plants microautophagy eliminates damaged chloroplasts and degrades cellular components during starvation [6]. Chaperone-mediated autophagy, which is facilitated by chaperones and does not involve membrane reorganization, directly targets nonessential proteins, which are transported through the lysosomal membrane for degradation [7]. Autophagy can be either selective or nonselective, and organelle-specific autophagy is also considered as selective autophagy [8]. Though there are several types of autophagy, the core autophagy components are conserved. This review will focus mainly on the role of core autophagy components in plant abiotic stress management.

## 2. Comparison of Plants and Yeast Core Autophagy Components

Over the past decade, the molecular and physiological understanding of plant autophagy has greatly increased. Most of the essential machinery required for autophagy seems to be conserved from yeast to plants [9,10,11]. In yeast, more than 40 autophagy-related genes (Atg) have been identified, half of which constitute the core autophagy machinery, including the Atg1–Atg13 kinase complex, the Atg9 cycling system, the phosphatidylinositol 3-kinase (PtdIns3K) complex and two ubiquitin-like conjugation systems (Atg8 and Atg12) [12]. According to the various available plant genome databases, the orthologs and paralogs of the yeast core autophagy components were identified in plants, ranging from lower species, such as algae, to major agricultural plants, such as cereals and vegetables [13,14,15,16,17,18,19]. Although there are differences among plant species, some components were unidentified in a regular BLAST search, but most of the core autophagy components were conserved. *Arabidopsis* is the model plant and has been extensively studied. *Oryza sativa* (rice), the most important crop in the world, has also been well studied. To compare the core autophagy components of plants and yeast, the following discussion is primarily based on the well-studied monocot plant rice and the dicot plant *Arabidopsis.*

The Atg1–Atg13 kinase complex is negatively regulated by targeting of rapamycin (TOR) in a nutrient-dependent manner. This complex includes five core components: Atg1, Atg13, Atg17, Atg29 and Atg31. Atg17 forms a complex with Atg29 and Atg31 under normal conditions and further interacts with Atg1 and Atg13 upon starvation to mediate preautophagosomal structure (PAS, a specific structure for autophagosome formation in yeast) organization [20,21]. Atg11 can then participate in this process by forming a scaffold with Atg17 for selective autophagy [22]. The PAS is a punctate structure proximal to the endoplasmic reticulum, which is the site where the Atg machinery assembles upon autophagy induction. A recent study showed that the PAS is a transient structure, and consists of a liquid-like condensate of Atg proteins formed through liquid–liquid phase separation [23]. Homologs of yeast *Atg1*, *Atg13* and *Atg11* have been identified in plants. There are three *Atg1* homologs in *Arabidopsis*, four in rice, and two *Atg13* homologs both in *Arabidopsis* and rice through alignment searching. In *Arabidopsis*, ATG1 and ATG13 can form a kinase complex facilitated by ATG11 under starvation conditions [24,25]. Two sensor kinases, SUCROSE NONFERMENTING1-RELATED PROTEIN KINASE1 (SnRK1) and TOR, have also been identified in *Arabidopsis* and rice. These two sensors can cooperatively perceive nutritional status to mediate ATG1–ATG13 kinase complex states and in turn activate or inhibit autophagy [26]. An ATG1-independent autophagy pathway exists in *Arabidopsis* under prolonged carbon starvation, in which the SnRK1 catalytic KIN10 subunit can directly phosphorylate the PtdIns3K complex ATG6 subunit to trigger autophagy [27]. These results showed that although the core autophagy mechanism is conserved in eukaryotic cells, plants need to engage multiple pathways to activate autophagy in response to the changeable ecological environment. A homolog of mammalian *mATG101* has also been identified in the *Arabidopsis* genome through DELTA-BLAST analysis, which is conserved in various eukaryotes, but not in *Saccharomyces cerevisiae*. It has also been shown to assemble into an active complex to initiate autophagy, but homologous or similar function genes to *Atg17*, *Atg19* and *Atg31* have not yet been found in plants (Figure 1a) [28]. The PAS is a specific structure involved in autophagosome formation in yeast. Recently, studies have confirmed the PAS is formed from liquid–liquid phase separation. Mammalian aggregaphy, the selective degradation of protein aggregates, also forms through liquid–liquid phase separation [29]. Whether liquid–liquid phase separation occurs in plant autophagy remains to be determined.

The PtdIns3K complex produces PI3P in the PAS or the ER to recruit downstream proteins. Yeast contains PtdIns3K complex I and PtdIns3K complex II. Only PtdIns3K complex I participates in autophagy, while PtdIns3K complex II functions in vacuolar protein sorting. The PtdIns3K complex I includes a class III PtdIns3K, vacuolar protein sorting protein 34 (Vps34); a serine/threonine kinase (Vps15); Vps30/Atg6; Atg14 and Atg38. Vps15 is required for the membrane association of Vps34. Atg14 is thought to recruit Vps34 and Vps30/Atg6 to PAS localization [30]. Atg38 helps to maintain the integrity of the PtdIns3K complex. A previous study identified that Atg38 can also interact with Atg8 via an ATG8-interacting motif (AIM), which is responsible for the PAS accumulation of the PtdIns3K complex [31]. Sequence homologs of these genes, except *ATG38,* have been found in plants. All of these genes appear to have only one copy except for *ATG14*, which has two copies in *Arabidopsis* [32]. However, the sequence homologs or functional similarity of *ATG14* was still missing in rice. In *Arabidopsis*, VPS34, VPS15 and ATG6 form the core complex, which can synthesize sufficient levels of PI3P [33,34,35]. This complex combines with ATG14 and VPS38 (an orthologue of the mammalian UV RADIATION RESISTANCE-ASSOCIATED GENE (UVRAG)) to better control the assembly of autophagosomes [32]. In yeast, Vps38 is responsible for endosome localization of the PtdIns3K complex II. In *Arabidopsis*, VSP38 is not only required for endosomal trafficking, but also promotes autophagy [36]. Whether VPS38 is involved in autophagy remains controversial [36,37], and the homolog of ATG38 remains to be identified (Figure 1b). 

Atg9 is an integral membrane protein that has been proposed to deliver lipids to autophagosomes and plays a critical role in phagophore nucleation and expansion in yeast [38]. Atg9 cycles between the PAS and non-PAS structures, which requires the help of Atg2, Atg8, Atg11, Atg23, Atg27, the Atg1-Atg13 complex and the PtdIns3K complex (Figure 1c). Under starvation conditions, Atg9 localizes to the PAS in an Atg23- and Atg27-dependent manner. Retrograde Atg9 transport from the PAS to cytoplasmic compartments depends on the Atg1–Atg13, Atg2–Atg18, and PtdIns3K complexes. Defects in any of the components of these complexes leads to the accumulation of Atg9 at the PAS [39]. Atg18 is a peripheral membrane protein that not only interacts with Atg9 but also binds to PI3P and PI(3,5)P2. Atg2, a ∼200 kDa peripheral membrane protein, is important for the PAS localization of Atg18 [40]. The results of a recent study suggest that Atg2 facilitates Atg18 binding to PI3P, and these two proteins act cooperatively to target the Atg2–Atg18 complex to Atg9 vesicles [40,41]. Atg2 may mediate autophagosome membrane expansion through its endoplasmic reticulum (ER) and pre-autophagosomal membrane location [38,40]. This previous work also provides evidence for the involvement of the ER in autophagosome formation in yeast. In addition, Atg2 can transfer lipids for Atg8 lipidation, and Atg18 can protect Atg8-PE from unregulated cleavage by Atg4 [42]. In *Arabidopsis*, *atg9* mutation causes the accumulation of abnormal autophagosome-related tubules upon autophagic induction. Immuno-electron microscopy (EM) and confocal microscopy analyses showed that these autophagosome-like structures were located near the rough ER. Three-dimensional tomographic reconstruction further revealed a direct connection between abnormal autophagosomal tubular structures and the ER in the *atg9-3* mutant [43]. All of these data demonstrate that ATG9 is required for the efficient budding of autophagosomes and that the original autophagosome is formed from the ER membrane. ATG9 also cycles between the ER and the cytoplasmic pool and regulates the trafficking of ATG18 on the autophagosomal membrane in a PI3P-dependent manner [43,44]. Although there is no PAS in plants, they have large and complex membrane systems, and the ER may not be the only source of autophagosome formation. ATG2 and ATG8 present multiple functions in plant development and abiotic stress responses, which will be discussed in detail in the next section. The sequence homologs of *Atg2, Atg9* and *Atg18* were also identified in rice. There is a single *ATG2* gene in rice like in *Arabidopsis*. In addition, there are two *ATG9* homologs in rice versus one in *Arabidopsis*. There is also a large *ATG18* family in rice including six numbers. Although the core ATG genes do exist in plants, plants have evolved either small or large families of specific ATG genes to perform specialized functions.

Two ubiquitin-like conjugation systems, Atg8-PE (phosphatidylethanolamine) and Atg12-Atg5-Atg16, are essential for autophagy progression. The lipidation of Atg8 requires the participation of a series of proteins. First, the C-terminus of Atg8 needs to be cleaved by a cysteine protease, Atg4, to expose a glycine residue. Then, it binds the E1-like enzyme Atg7 and is transferred to the E2-like enzyme Atg3. Finally, Atg8 is covalently linked to the membrane lipid phosphatidylethanolamine (PE) to form the Atg8-PE complex [45]. The conjugation between Atg8 and PE is reversible. Atg8 can also be released from the membrane lipid by the deconjugating enzyme function of Atg4. The N-terminal motif of Atg4, close to the catalytic motif, plays a key role in specific Atg8 deconjugation [46]. Atg12 and Atg8 are both ubiquitin-like proteins. The activity of Atg12 also requires the E1-like enzyme Atg7. Atg12 is then transferred to another E2-like enzyme, Atg10, and is finally conjugated to Atg5. The Atg12-Atg5 conjugate further interacts with a coiled-coil protein, Atg16, to form a tetrameric Atg12-Atg5-Atg16 complex. Atg16 interacts with Atg21 in a PI3P-dependent manner to recruit this complex to the PAS [47,48]. The two conjugation systems are closely related, as Atg12-Atg5-Atg16 is necessary for Atg8-PE formation and lipidation site determination [39,49]. In addition, Atg12 can interact with the Atg1 kinase complex, which serves as a scaffold for PAS organization [40]. These two ubiquitin-like conjugation systems have been well studied in plants, especially the ATG8 system, which also needs to be cleaved by ATG4 to finally form the ATG8-PE lipid complex [50,51,52]. The ATG12-ATG5 complex is essential for ATG8-mediated autophagy in plants by promoting ATG8 lipidation [53,54]. The homolog of ATG16 also exists in plants, but its specific function in autophagy remains to be determined (Figure 1d). ATG8 has been analyzed in more detail. There are nine homologs in *Arabidopsis* and five homologs in rice. Different members show distinct expression patterns and may have distinct functions during a plant’s development or its response to different stress conditions [55]. Almost all autophagy-related receptors/adaptors identified to date have ATG8 interaction domains (AIMs/UIMs) or can directly interact with ATG8. These results imply sophisticated roles of ATG8 in autophagy and plant development or stress response, which were illustrated in detail in recent reviews [55,56,57]. 

Through the above steps, the autophagosome is filled with cargos. The mature autophagosome then fuses with and delivers its contents to the vacuole. The contents are then broken down by various hydrolases into carbohydrates, amino acids and lipids that are returned to the cytosol and recycled to synthesize new products or reused for other purposes. The study of yeast autophagy has provided a solid foundation for autophagy-related research in plants. Through homologous comparison and mutants analysis, many functional autophagy genes have been identified in plants. By screening autophagy-related interacting proteins, plant-specific genes required for autophagy were also identified, most of which are induced by biotic and abiotic stresses. Autophagy is induced by starvation and senescence, which is well established and is not included in this review. Abiotic stresses, including drought, salt, heat and cold, are inevitable environmental stresses encountered by plants. In recent years, an increasing number of studies have shown that autophagy plays an important role in plant stress management. In this review, we will focus on the role of autophagy in plant abiotic stress regulation in the order of autophagosome formation.

## 3. The Role of Core Autophagy Components in Plant Abiotic Stress 

Adverse environmental conditions negatively affect agricultural production and reduce crop yield, both qualitatively and quantitatively, and accompany significant transcriptomic changes in plant cells [58,59,60,61,62,63,64,65]. Abiotic stresses, including oxidative, osmotic, drought, salt, cold and heat stresses, can not only influence *ATG* genes expression but also induce autophagosome formation in plant cells. Autophagy is an important stress response mechanism in plants. Homology-based analyses have identified conserved *ATG* genes in plants, some of which have been reported to participate in the plant abiotic stress response.

In plants, the homologs of the Atg1–Atg13 complex, the PtdIns3K complex, and the Atg2–Atg8 complex have been identified. Only ATG1, ATG2, ATG13 and ATG18 have been found to be needed for ATG9 cycles. ATG9, ATG11 and PI3K act upstream of ATG2 [25,43,66,67]. Their functions have been characterized in detail, and most are related to plant abiotic stress responses (not including starvation). There are four homologs of *Atg1* in *Arabidopsis*. KIN10, an *Arabidopsis* ortholog of mammalian AMPK, can activate autophagy in response to drought and hypoxic stress by affecting ATG1 phosphorylation [68]. The PtdIns3K complex is also conserved in *Arabidopsis*. The homologs of *Vps34*, *Atg6* and *Vps15* are all present with only one copy, while *Atg14* appears to have two homologs. The core components of the PtdIns3K complex are involved in plants abiotic stress regulation. *Arabidopsis* PI3K plays a positive role in salt tolerance. The *pi3K* mutants and wortmannin-treated plants exhibit an overly salt-sensitive phenotype [69]. PI3K can facilitate the internalization of plasma membrane intrinsic protein 2;1 (PIP2;1) from the plasma membrane (PM) into the vacuole under salt stress to decrease root water permeability [70]. In rice, there are three *Atg6* homologs, which show differential expression when subjected to heat, cold and drought stress. *OsATG6a* is up-regulated by drought but down-regulated by HS. *OsATG6b* is up-regulated by drought but down-regulated by cold stress. *OsATG6c* is up-regulated by all examined stresses, including heat, drought and cold [71]. In barley, the expression of *HvATG6* is also up-regulated by various abiotic stresses, including H_2_O_2_ treatment, high salinity, drought and low temperature. Knockdown of *HvATG6* in barley leaves through barley strip mosaic virus (BSMV)-induced gene silencing leads to accelerated yellowing under H_2_O_2_ treatment [72]. These results show that the stress-specific response of *ATG6* will help plants better deal with different stress conditions, but the biological mechanisms and action of these genes under abiotic stresses remain unclear. Although autophagy normally plays a positive role in plant abiotic stress regulation, not all of its components are up-regulated under stress conditions, and some can also be inhibited, which may be related to the extensive functions of autophagy and the extent of plant damage.

*Arabidopsis* has a single *ATG2* gene that is expressed ubiquitously throughout the plant. The *atg2* knockout mutants display typical autophagy-defective phenotypes during senescence and stress conditions [73,74]. The T-DNA insertion *Arabidopsis* mutant *atg2-5* shows impaired low-CO_2_-induced stomatal opening. Plants can regulate stomatal opening and closure to cope with diverse environmental stresses [75]. In studying this mutant, the author found that autophagy controls guard cell reactive oxygen species (ROS) homeostasis by eliminating oxidized peroxisomes, thereby allowing stomatal opening. The disruption of other autophagy genes in *Arabidopsis*, including *ATG5*, *ATG7*, *ATG10* and *ATG12*, causes similar stomatal defects and results in the constitutive accumulation of high ROS levels in guard cells [76]. These results illustrate that autophagy can alter ROS levels to regulate stomatal states. Moreover, *ATG2* is induced by high temperatures in *Arabidopsis*. The fresh weight and chlorophyll retention of *atg2-1* mutants are dramatically decreased compared with those of wild-type (WT) plants under high-temperature conditions. The mRNA levels of *ATG5, ATG6, ATG12A* and *ATG18A* also significantly increase in WT plants under high-temperature conditions [74]. In addition, autophagy plays a role in resetting the cellular memory of heat stress (HS). Autophagy is induced by thermopriming and remains at a high level long after stress termination. The autophagy-deficient mutants *atg2-1, atg5-1, atg12ab* and *atg18a-2* show significantly better survival under postmemory HS than WT plants, suggesting that autophagy negatively controls thermomemory in *Arabidopsis*. Upon further investigation, the author reported that the autophagy mutants retain heat shock proteins longer than the WT and concomitantly display improved thermomemory. These results show that autophagy mediates the specific degradation of heat shock proteins at later stages of the thermorecovery phase, leading to compromised heat tolerance after the second HS [77]. CaATG6 was predicted to interact with CaHSP90 family members and is related to heat stress tolerance in pepper [78]. These link autophagy to heat stress through regulation heat shock protein levels. In the presence of low NaCl concentrations, *Arabidopsis atg9* and *atg2* mutants germinated faster than the WT, while the *atg5* and *atg7* mutants showed the opposite behavior. In the presence of higher NaCl concentrations, germination slowed down in all lines [79]. These results demonstrate that under different degrees of stress, autophagy has different manifestations. The ATG18 proteins are a large protein family in plants, similar to that found in mammals but not in yeast. There are eight *Atg18* homologs in *Arabidopsis* and six in rice. In *Arabidopsis*, only *ATG18A* is up-regulated by both salt and osmotic stresses. *RNAi-ATG18A Arabidopsis* plants are more sensitive to salt and osmotic treatment. All of these results show that ATG18A plays a positive role in salt and osmotic stress regulation in *Arabidopsis*. Furthermore, *RNAi-ATG18A Arabidopsis* seedlings are hypersensitive to oxidative stress and exhibit delayed growth and severe bleaching relative to WT *Arabidopsis*. Following treatment with methyl viologen (MV), an inducer of ROS, autophagosome accumulation is observed in WT *Arabidopsis* plants but not in *RNAi-ATG18A Arabidopsis* plants, and *RNAi-ATG18A Arabidopsis* plants accumulate higher levels of oxidized proteins than WT *Arabidopsis* plants [80,81]. These results show that autophagy plays a role in the clearance of oxidized proteins following oxidative stress in *Arabidopsis*. Inhibiting ROS production also has an effect on autophagy induction under salt stress conditions but has no effect under osmotic stress. These results illustrate that ROS can link autophagy and abiotic stress by acting as signaling molecules. In addition, *ATG18A* overexpression can improve transgenic plants resistance to drought stress in both tomato and apple plants [82,83]. Large numbers of *ATG18* homologs exist in plants, and different numbers of these homologs respond to different stresses, demonstrating that plants have evolved complex autophagy mechanisms to improve their survival in extreme environments. 

The two ubiquitin-like conjugation systems are also well conserved in plants. Plants contains more members of the protein family involved in these systems. There are nine *ATG8* family numbers in *Arabidopsis* and five *ATG8* family numbers in rice versus one in yeast. In *Arabidopsis*, all nine *ATG8s* are expressed throughout the plant in distinct expression patterns, implying that each member may have a distinct function during development or under various stress conditions; however, only some of these genes have been reported to be involved in abiotic stress responses. The overexpression of a *GFP-ATG8F-HA* fusion protein was found to exert a considerably stronger negative effect on the growth of *Arabidopsis* plants under mild NaCl treatment with increased electrolyte leakage, indicating greater damage to the cell membrane system than was observed in the control seedlings. The effect of the expression of the *GFP-ATG8F-HA* construct on the response of the plants to relatively mild osmotic stresses is the same as that of salt [84]; however, *Arabidopsis* overexpressing *ATG8A* performed better than WT plants in germination assays on NaCl-containing plates. In addition, ATG8A and ATG8E can serve as autophagy markers. When plants were subjected to different abiotic stresses, GFP-ATG8A or ATG8E-GFP accumulated in autophagosomes, demonstrating that ATG8 participates in plant abiotic stress regulation [77,79,81]. The different ATG8 family members associated with the responses to different stresses allow fine-tuning of the regulatory mechanism of autophagy [85]. In wild emmer wheat, the expression of *TdATG8* is also strongly induced by drought and osmotic stress, especially in the roots relative to the leaves [86]. There are also two/nine members, respectively, of the ATG4/ATG8 families in common wheat (*Triticum aestivum* L.). The expression profiles of *TaATG4a, 4b, 8a, 8g* and *8h* all show up-regulated expression upon exposure to high salinity, drought and low temperature [87]. Two other ubiquitin-like conjugation system components, ATG5 and ATG7, have been reported to respond to salt and oxidative stress in *Arabidopsis*. *ATG5*- or *ATG7*-overexpressing plants exhibited increased resistance to oxidative stress, delayed ageing and enhanced growth, and plants with mutations in these genes showed the opposite behavior [88]. In addition, both *Arabidopsis* and tomato plants with mutations in these two genes are more sensitive to HS than WT plants [77,89]. Atg10, an E2-like enzyme, is another important component of the Atg12-Atg5-Atg16 ubiquitin-like conjugation systems. There is one *Atg10* homologue in *Arabidopsis*, but there are two *Atg10* homologs in rice, *OsATG10a* and *OsATG10b*. Only *osatg10b* mutants are sensitive to high salt and methyl viologen treatment, resulting in the formation of fewer autophagosomes than that observed in WT [90]. In addition, overexpression of the autophagy-related gene *MdATG10* in apples can increase autophagic activity in the roots and enhance transgenic plants salt tolerance [91]. These results demonstrate that autophagy plays an important role in the survival of plant cells under different abiotic stresses. Importantly, *ATG* genes can quickly respond to different abiotic stress conditions in the initiation phase of autophagy.

Autophagy is also involved in the regulation of abiotic stress in other plants. Autophagy has been reported to be involved in pepper (*Capsicum annuum* L.) tolerance to abiotic stresses. The author identified 15 core ATG members, including 29 ATG proteins with corresponding conserved functional domains, in the whole genome of pepper via the HMM method. Under salt, drought, heat and cold stresses, the expression levels of *CaATG* genes changed in a stress type-dependent pattern, and the accumulation of autophagosome puncta increased. These results indicate the linkage of autophagy in the response of pepper to abiotic stresses [78]. *Camellia sinensis* autophagy-related genes (*CsARGs*) also respond to abiotic stress and most *CsARGs* were upregulated at different time points during the abiotic stress treatment [17]. Autophagy-related genes were also identified in *Citrus sinensis.* Most of the *CsATGs* were significantly changed in response to drought, cold, heat, salt and mannitol treatment. In addition, ectopically expressed *CsATG18a* and *CsATG18b* in *Arabidopsis* showed enhanced tolerance to osmotic, salt, drought (*CsATG18a*) or cold (*CsATG18b*) stress, compared to wild-type plants [14]. Drought stress also up-regulates the expression of the autophagy-related genes *ATG1, ATG8, ATG9* and *ATG12* in *Caragana korshinskii* [15]. Homologs of *Atg3* from apples are reported to improve transgenic *Arabidopsis* salt and osmotic stress resistance [92]. The above findings demonstrate that the core autophagy components are conserved among plants and play a wide range of roles in plant abiotic stress management. 

## 4. The Role of Selected Autophagy in Plant Abiotic Stress

Autophagy was initially defined as a bulk degradation process that causes massive degradation of cellular components; however, in recent years, cumulative evidence has indicated that the recruitment of cargo to autophagosomes is highly selective [56,93]. Several selective autophagy receptors have been characterized in plants. These selective autophagy receptors link organelles, protein aggregates or other cargo to the autophagy machinery by binding to both the fated cargo and ATG8 through conserved ATG8-interacting motif (AIM) or ubiquitin-interacting motif (UIM). Several characterized autophagy receptors function in the plant abiotic stress response (Table 1).

The next to BRCA1 gene 1 (NBR1) is a functional hybrid protein of the mammalian autophagy receptor p62 (also known as Sequestosome1/SQSTM1) and a neighbor of BRCA1 (NBR1) that specifically targets stress-induced, ubiquitinated protein aggregates [94,95]. Both p62 and NBR1 preferentially target K63-linked polyubiquitylated proteins and mediate their aggregation and autophagic clearance in an LC3-interacting region (LIR)- and UBA-dependent manner. NBR1 and p62 oligomerize but can also function independently [96]. *Arabidopsis* AtNBR1 can bind ATG8 via the AIM motif and ubiquitinate proteins via the ubiquitin-associated domain [97]. The expression of *AtNBR1* is upregulated under HS in *Arabidopsis* [98]. Moreover, under HS conditions, *atnbr1* mutants accumulate more puncta in the cytoplasm compared with WT. GFP-NBR1 puncta accumulate in WT plants but not in *atg7* mutants under HS conditions. During the HS recovery phase, more NBR1 puncta accumulate in WT plants, and the NBR1 protein accumulates at substantially higher levels in *atg5-1* and *atg18a-2* mutants than in WT plants [98,99]. These findings demonstrate that NBR1 puncta formation is autophagy dependent and that NBR1 is required not only for the heat-induced formation of autophagosomes but also for the degradation of substrates in an autophagy-dependent pathway throughout the HS stage. Further analysis showed that NBR1 plays a crucial role as a receptor for the selective autophagy-mediated degradation of heat shock protein 90.1 (HSP90.1) and rotamase FKBP1 (ROF1) during recovery from HS to regulate HS memory in *Arabidopsis* [100]. In addition, *nbr1* mutants are hypersensitive to oxidative, drought and salt stress relative to WT plants [78,95,97,98]. Similar results were observed in tomato plants in which *NBR1* was silenced by VIGS. *ATGs* and *NBR1* gene silencing triggered the accumulation of ubiquitinated insoluble proteins and decreased the number of autophagosomes under cold and heat stress [88,101]. In poplars, *PagNBR1* is also induced by salt stress. *PagNBR1* overexpressing poplars displayed more salt stress tolerance by accelerating antioxidant system activity and autophagy activity [102]. These results demonstrate the important roles of NBR1 in resisting stress conditions via the autophagy pathway.

Atg8-Interacting Proteins 1/2/3 (ATI1/2/3) are AIM-motif-containing proteins identified through a yeast two-hybrid screen for proteins interacting with ATG8. ATI1 and ATI2 are homologous in that each contain two AIM motifs and a transmembrane domain. These two proteins define a newly identified stress-induced compartment that moves along the ER network and is subsequently transported to the vacuole in *Arabidopsis* plants [103,104]. Salt stress promotes ATI1 protein accumulation. *Arabidopsis* that are deficient in both homologs (*ATI-KD*) display increased sensitivity to salt treatment both at the seedling stage and in older plants but no effect was observed on germination. The authors found that ATI1 may play a role in the elimination of damaged plastid and ER proteins produced during salt stress [105]. ATI3 proteins contain a WxxL motif at the C-terminus required for ATG8 interaction. ATI3 homologs are found in dicots but not in other organisms, including monocots. The *ati3* mutant plants display hypersensitivity to HS, and the interaction of ATI3 and ATG8 is increased under HS [106]. 

Dominant suppressor of Kar 2 (DSK2), a ubiquitin-binding receptor, is another ATG8-interacting protein with an AIM motif. *DSK2-RNAi Arabidopsis* plants display increased sensitivity to drought stress and increased levels of BES1 (BRI1-EMS SUPPRESSOR 1) relative to the WT. BES1 is a master regulator of the brassinosteroid (BR) pathway. DSK2 can be phosphorylated by another negative regulator of the BR pathway, BIN2, which promotes DSK2-ATG8 interaction [107]. These results suggest that DSK2, acting as an autophagy receptor, specifically directs BES1 degradation through the autophagy pathway under drought stress conditions. This link between BR signaling and autophagy means that these processes regulate the plant stress response together. Phytohormones play important roles in plant growth, development, abiotic and biotic stress responses. Phytohormone signals and the autophagy pathway jointly regulate plant responses to abiotic stress [108]; however, the relationship between phytohormones and autophagy in plant abiotic stress regulation remains unclear.

Tryptophan-rich sensory protein/translocator (TSPO) is a type of tryptophan-rich sensory protein/peripheral-type benzodiazepine receptor (TspO/MBR) domain-containing membrane protein, which also has an AIM motif that interacts with ATG8 [109]. The expression of *AtTSPO* is induced by osmotic and salt stress [110,111]. *AtTSPO* overexpression makes plants hypersensitive to salt stress, possibly because AtTSPO can bind the plasma membrane aquaporin AtPIP2;7 and regulate its degradation through the autophagy pathway, and the overdegradation of aquaporins impairs cell water status [112]. In addition, similar to NBR1, TSPO is degraded via the autophagy pathway in a manner dependent on ATG5 and the PI3K complex [109]. Another protein from *Medicago*, cold acclimation-specific 31 (MtCas31), can also regulate MtPIP2;7 stability through the autophagy pathway and participate in drought stress regulation. MtCAS31 directly interacts with MtATG8a in the AIM-like motifs YXXXI and MtPIP2;7, supporting its function in autophagic degradation. The overexpression of *MtCAS31* promotes autophagy and MtPIP2;7 degradation under drought stress. These results demonstrate that MtCAS31 functions as a positive regulator of drought stress in *Medicago* and participates in the drought-induced autophagic degradation of MtPIP2;7 as a cargo receptor [113]. The selected substrates of AtTSPO and MtCAS31 are from the same family of proteins but do not overlap, which may be related to the meticulous regulation of autophagy.

Constitutively stressed 1 (COST1), a plant-specific gene, can also interact with ATG8E but does not have a typical AIM or UIM motif. The *cost1* mutant exhibits decreased growth and increased drought tolerance, together with constitutive autophagy and increased expression of drought response genes. *COST1* overexpression confers drought hypersensitivity and reduces autophagy. COST1 co-localizes with ATG8E and NBR1 in autophagosomes and directly affects the ATG8E protein level, indicating that it plays a pivotal role in the direct regulation of autophagy [114,115]. The above results illustrate that COST1 is a negative regulator of both drought resistance and autophagy. The increased drought tolerance of the *cost1* mutant is due to autophagy activation [114,115,116]. The majority of the autophagy receptors/adaptors identified to date are ATG8-interacting proteins and thus ATG8 is an essential regulator of autophagy; however, the regulation of autophagy is complicated, and autophagy plays a role in almost every part of the plant life cycle. ATG8-independent receptors/adaptors must exist and require further exploration.

## 5. Conclusions and Prospects

Due to their sessile nature, plants are more vulnerable to abiotic stress than mobile organisms. These stresses challenge agriculture and food security globally. Plants have evolved many systems to alleviate abiotic stress. Autophagy is one of the most important and effective ways to degrade harmful proteins/damaged organs and recycle products/reuse these materials; however, most research on plant autophagy is based on yeast and mammals. Although the core autophagy pathway is conserved in plants, a few genes have yet to be identified in plant genomes, and some genes have expanded into gene families. Do other novel autophagy genes remain to be discovered in plants? Do the genes within families show functional differentiation? These questions warrant future investigation. In recent years, many studies on selected autophagy pathways have emerged, and some plant-specific cargo receptors and conserved cargo receptors have been identified. However, the molecular mechanisms of selective autophagy pathways still need further elucidation. Autophagy can respond to many abiotic stresses, and the same *ATG* gene can respond to different abiotic stresses. Does abiotic stress exhibit crosstalk in an autophagy-dependent pathway? Reported studies have increased the complexity of the known plant autophagy system. How these studies can be applied to cultivate more stress-resistant and high-yield crops must be given more attention.

## Figures and Tables

**Figure 1 ijms-22-04075-f001:**
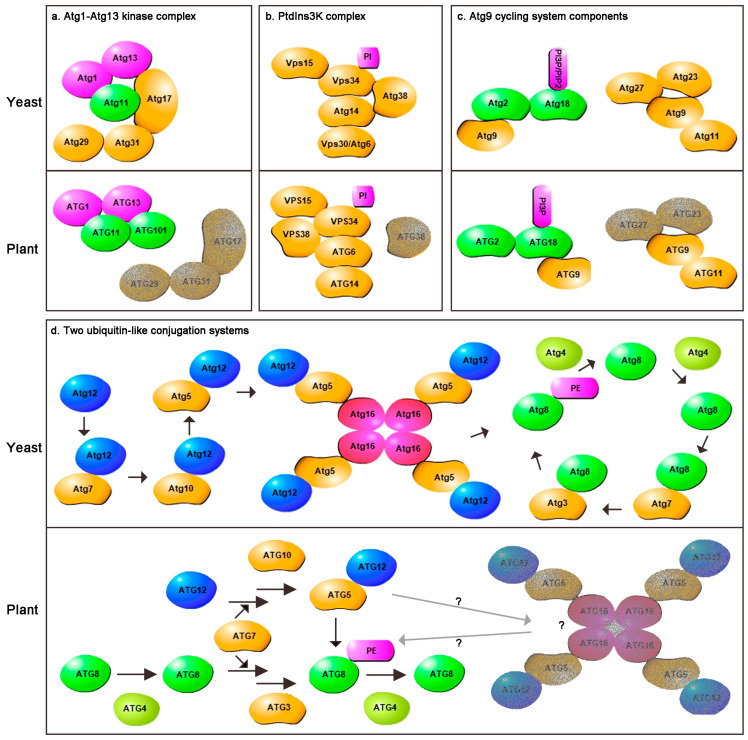
The comparison of yeast and *Arabidopsis* core autophagy components. (**a**) Atg1–Atg13 kinase complex. (**b**) The phosphatidylinositol 3-kinase (PtdIns3K) complex. (**c**) The Atg9 cycling system components. (**d**) The two ubiquitin-like conjugation systems. The unidentified components or complexes in plant are covered by grey.

**Table 1 ijms-22-04075-t001:** The cargo receptors/adapters in plant abiotic stress responses.

Plants.	Features	Involve in
AtNBR1/AT4G24690	AIM motif	Heat, oxidative, drought, salt, cold
SlNBR1a/ Sl03g112230 SlNBR1b/ Sl06g071770	AIM motif	Heat, cold
AtATI1/AT2G45980	AIM motif	Salt
AtATI2/AT4G00355	AIM motif	Salt
AtATI3A/AT1G17780 AtATI3B/AT2G16575 AtATI3C/AT1G73130	AIM motif	Heat
AtDSK2A/AT2G17190 AtDSK2B/AT2G17200	AIM motif	Drought
AtTSPO/At2g47770	AIM motif	Osmotic, salt
AtCOST1/AT2G45260	--	Drought
MtCAS31/EU139871	AIM-like motifs	Drought

--, not present or not identifiable.

## Data Availability

Please refer to suggested Data Availability Statements in section “MDPI Research Data Policies” at https://www.mdpi.com/ethics.

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
