# Peer review of "Autophagy in Plant Abiotic Stress Management"

_ijms, 2021, doi:10.3390/ijms22084075_

Round 1
Reviewer 1 Report
Dear Authors,
The topic you target is very important for journal readers. Unfortunately your review does not adress the topic accurately for many reasons detailed below:
- The title of the manuscript is misleading : you claim you are covering plant kingdom but you mainly speak abour Arabisdopsis. The report of rice and barley was only on page 6, wheat on page 7, Capsicum on page 8 ....
- There is no definition of Autophagy in the beginning of the manuscript. It is worth to give a definition for readers that are not familiar with autopphagy. The only definition i found was on page 8 line 311.
- You use the word autophagy for macroautophagy : I not agree with that and think this misleading to readers. You need to use macroautophagy.
- Please be precise when you say in plants : which plant? it is crucial to do so this is reserach andf we need to be very precise.
- You was using two models yeast and mammals but no comparaison of yeast, mamals and plant components is provided! Mammals which mammals ? please be precise this is misleading you really need a major major reorganisation of data !
- Please define more widely PAS and non PAS structures. I feel your manuscript can not be read by scientists not familiar with autophagy or students.
- I feel that adding a figure that represents interaction of different autophagy genes in yeast, mammals and plants would be better than writing pages of text. Please provide a complete figure of autophagy genes interactions.
- No description of autophagy components in mammals in the text.
- In Table 1 only Arabidopsis genes are reported why not rice, wheat, barley, ...since autophagy genes have also been described in plants and these species represent better plants that the weed Arabidoopsis.
- Please define precisely what you mean by homolog: is it defined by sequence similarity? functional similarity? activity? Please be precise !!!
- Table 1 isd adapted from Noboru Mizushima [49] so where is your input and the input of your new review or should we conclude that the you not provide any progress beyond the work of Mizushima?
- Please discuss all aspects in all plant species where studies have been reported ! you claim you are reviewing I not feel it is a review ! I feel it is review of autophagy in Arabidopsis with minor examples of stuidies in other plants.
Well I hope you a good luck to adress all these aspects and looks for reading a nice comprehensive review that I can suggest its acceptance and that can help researchers and journal redaers to have a comprehensive view of autophagy in PLANTS.
Best regards
Author Response
Dear reviewer,
On behalf of my co-authors, we thank you very much for your letter and for the reviewers’ comments concerning our manuscript entitled “Autophagy and plant abiotic stress management” (ijms-1168721). Those comments are all valuable and very helpful for revising and improving our paper, as well as the important guiding significance to our researches. We have studied comments carefully and have made correction which we hope meet with approval. We have carefully evaluated the reviewers’ critical comments and thoughtful suggestions, responded to these suggestions point-by-point, and revised the manuscript accordingly. All changes made to the text are used red font instead of black so that you may be easily identified.
The main corrections in the paper and the responds to the reviewer’s comments are as flowing:
- The title of the manuscript is misleading : you claim you are covering plant kingdom but you mainly speak abour Arabisdopsis. The report of rice and barley was only on page 6, wheat on page 7, Capsicum on page 8 ....
Thank you for reviewer’s suggestion. We add rice as another major compare plant in the whole paper (line48-55 and so on). And make specific illustrate about autophagy in the regulation of other plants including barley (line219-222), tomato and apple (line273-274), wild emmer wheat (line296), common wheat (298), pepper (line317), Camellia sinensis and Citrus sinensis (line 324-329) to abiotic stress. Hope to meet the requirement.
- There is no definition of Autophagy in the beginning of the manuscript. It is worth to give a definition for readers that are not familiar with autopphagy. The only definition i found was on page 8 line 311.
Considering the Reviewer’s suggestion, we defined Autophagy in the beginning of the manuscript included abstract (Line6-8 and line18-20).
- You use the word autophagy for macroautophagy : I not agree with that and think this misleading to readers. You need to use macroautophagy.
We feel very sorry that we haven't made it clear. Though there are several types of autophagy, the core autophagy components are conserved. This review will focus mainly on the role of core autophagy components in plants abiotic stress management. We have made change in line 38-40.
- Please be precise when you say in plants : which plant? it is crucial to do so this is reserach andf we need to be very precise.
We are sorry about our inaccurate description. We have change plants to specific species (line125, 170…)
- You was using two models yeast and mammals but no comparaison of yeast, mamals and plant components is provided! Mammals which mammals ? please be precise this is misleading you really need a major major reorganisation of data !
We feel very sorry that we haven't made it clear. We only use yeast as model.
Mammals are mentioned in this article because ATG101 is not identified in yeast. And mammals mentioned in the paper refers to human.
- Please define more widely PAS and non PAS structures. I feel your manuscript can not be read by scientists not familiar with autophagy or students.
Considering the Reviewer’s suggestion, we defined PAS in the manuscript (Line63-64). We hope make it clear.
- I feel that adding a figure that represents interaction of different autophagy genes in yeast, mammals and plants would be better than writing pages of text. Please provide a complete figure of autophagy genes interactions.
According to the reviewers’ comments, we add figure1 to elucidate autophagy genes interactions. Hope to meet the requirement.
- No description of autophagy components in mammals in the text.
We feel very sorry that we haven't made it clear. We only use yeast as model. And made the corresponding change in line 43-44.
- In Table 1 only Arabidopsis genes are reported why not rice, wheat, barley, ...since autophagy genes have also been described in plants and these species represent better plants that the weed Arabidoopsis.
Considering the Reviewer’s suggestion, we changed Table 1 to Figure1. Hope to meet the requirement.
- Please define precisely what you mean by homolog: is it defined by sequence similarity? functional similarity? activity? Please be precise !!!
According to the reviewers’ comments, we changed homolog to sequence homologs in line 24, 100, 138…
- Table 1 isd adapted from Noboru Mizushima [49] so where is your input and the input of your new review or should we conclude that the you not provide any progress beyond the work of Mizushima?
Considering the Reviewer’s suggestion, we changed Table 1 to Figure1. Hope to meet the requirement.
- Please discuss all aspects in all plant species where studies have been reported ! you claim you are reviewing I not feel it is a review ! I feel it is review of autophagy in Arabidopsis with minor examples of stuidies in other plants.
Considering the Reviewer’s suggestion, we add rice as another major compare plant in the whole paper (line48-55 and so on). And make specific illustrate about autophagy in the regulation of other plants including barley (line219-222), tomato and apple (line273-274), wild emmer wheat (line296), common wheat (298), pepper (line317), Camellia sinensis and Citrus sinensis (line 324-329) abiotic stress. Hope to meet the requirement.

Reviewer 2 Report
The paper was focused on the topic of autophagy and plant abiotic stress management. Throughout their life cycle, plants need to exchange material, energy and information with the outside world. Autophagy is an evolutionarily conserved process that occurs ubiquitously in all eukaryotic cells and plays many physiological roles in maintaining cellular homeostasis. In recent years, an increasing number of studies have shown that autophagy can be induced not only by starvation but also as a cellular response to various abiotic stresses, including oxidative, salt, drought, cold and heat stresses.
The manuscript is correctly written and organized. However, it needs significant improvement in English style and grammar.
In addition, I recommend adding few references of newly published articles presenting the results of global changes at transcriptomic and microRNA levels in plants subjected to abiotic stress. Especially, results regarding next generation sequencing (RNA-seq) of plant samples would significantly increase the scientific value of the manuscript.
Authors should prepare some colourful diagrams, charts, and figures. Graphic material will increase the readability of the article and it will be more interesting for readers.
Author Response
Dear reviewer,
On behalf of my co-authors, we thank you very much for your letter and for the reviewers’ comments concerning our manuscript entitled “Autophagy and plant abiotic stress management” (ijms-1168721). Those comments are all valuable and very helpful for revising and improving our paper, as well as the important guiding significance to our researches. We have studied comments carefully and have made correction which we hope meet with approval. We have carefully evaluated the reviewers’ critical comments and thoughtful suggestions, responded to these suggestions point-by-point, and revised the manuscript accordingly. All changes made to the text are used red font instead of black so that you may be easily identified.
The main corrections in the paper and the responds to the reviewer’s comments are as flowing:
- The manuscript is correctly written and organized. However, it needs significant improvement in English style and grammar.
Considering the Reviewer’s suggestion, we have revised our English grammar and awkward phrasing by AJE polishing.
- In addition, I recommend adding few references of newly published articles presenting the results of global changes at transcriptomic and microRNA levels in plants subjected to abiotic stress. Especially, results regarding next generation sequencing (RNA-seq) of plant samples would significantly increase the scientific value of the manuscript.
Considering the reviewer’s valuable proposal, we add some next generation sequencing of plant article in manuscript (line192-194 and line 677-692). Hope to meet the requirement.
- Yan F, Zhu Y, Zhao Y, Wang Y, Li J, Wang Q, Liu Y: De novo transcriptome sequencing and analysis of salt-, alkali-, and drought-responsive genes in Sophora alopecuroides. BMC genomics. 2020, 21:423.
- Arisha MH, Aboelnasr H, Ahmad MQ, Liu Y, Tang W, Gao R, Yan H, Kou M, Wang X, Zhang Y et al: Transcriptome sequencing and whole genome expression profiling of hexaploid sweetpotato under salt stress. BMC genomics. 2020, 21:197.
- Pan L, Yu X, Shao J, Liu Z, Gao T, Zheng Y, Zeng C, Liang C, Chen C: Transcriptomic profiling and analysis of differentially expressed genes in asparagus bean (Vigna unguiculata ssp. sesquipedalis) under salt stress. PloS one. 2019, 14:e0219799.
- Sun M, Huang D, Zhang A, Khan I, Yan H, Wang X, Zhang X, Zhang J, Huang L: Transcriptome analysis of heat stress and drought stress in pearl millet based on Pacbio full-length transcriptome sequencing. BMC Plant Biol. 2020, 20:323.
- Kumar RR, Goswami S, Sharma SK, Kala YK, Rai GK, Mishra DC, Grover M, Singh GP, Pathak H, Rai A et al: Harnessing Next Generation Sequencing in Climate Change: RNA-Seq Analysis of Heat Stress-Responsive Genes in Wheat (Triticum aestivum L.). OMICS. 2015, 19:632-647.
- Abdelrahman M, Jogaiah S, Burritt DJ, Tran LP: Legume genetic resources and transcriptome dynamics under abiotic stress conditions. Plant Cell Environ. 2018, 41:1972-1983.
- He J, Jiang Z, Gao L, You C, Ma X, Wang X, Xu X, Mo B, Chen X, Liu L: Genome-Wide Transcript and Small RNA Profiling Reveals Transcriptomic Responses to Heat Stress. Plant Physiol. 2019, 181:609-629.
- Bashir K, Matsui A, Rasheed S, Seki M: Recent advances in the characterization of plant transcriptomes in response to drought, salinity, heat, and cold stress. F1000Res. 2019, 8.
- Authors should prepare some colourful diagrams, charts, and figures. Graphic material will increase the readability of the article and it will be more interesting for readers.
Considering the Reviewer’s suggestion, we changed Table 1 to Figure1. Hope to meet the requirement.

Round 2
Reviewer 1 Report
I am happy the authors considered my recommendations. I feel the paper is ready for publication and meets IJMS standards. I suggest acceptance and rapid publication